# Efficacy of home-based and inpatient treatment for children and adolescents in psychiatric crisis: A systematic review protocol

Karolina Foremnik[1]*, Gaby Sroczynski[2], Jan Stratil[2], Anja Neumann[1], Barbara Buchberger[1,3]

1 Faculty of Medicine, University of Duisburg-Essen, Essen, Germany, 2 UMIT TIROL - University for Health Sciences and Technology, Institute of Public Health, Medical Decision Making und HTA, Hall in Tirol, Austria, 3 Robert Koch Institute, Berlin, Germany

* karolina.foremnik@uni-due.de

## Abstract

### Purpose

Children and adolescents experiencing psychiatric crises often undergo inpatient treatment, which may limit family involvement, stigmatize young individuals, and impede the application of therapeutic outcomes in their daily lives. This situation can result in increased rates of rehospitalization, the development of chronic conditions, and prolonged hospital stays. Home-based treatment represents a potential alternative to traditional inpatient care. The objective of the planned systematic review is to provide a comprehensive comparison of the effectiveness and cost-effectiveness of inpatient and home-based treatment modalities, with a particular focus on primary outcome parameters such as psychopathology, family functioning, and social functioning. Furthermore, secondary outcomes, including rates of relapses and rehospitalizations, will be evaluated.

### Methods

The systematic search will be conducted using Medline, Embase, PsycInfo and Cochrane databases, following the guidelines of the Preferred Reporting Items for Systematic Reviews and Meta-Analyses (PRISMA). The included studies will undergo a rigorous quality assessment using the Cochrane Risk of Bias (RoB2) tool for randomized trials and the Risk of Bias in Non-randomized Studies - of Interventions (ROBINS-I) tool for non-randomized studies. Where appropriate, data will be synthesized by meta-analysis using R-Studio and supplemented by sensitivity analyses to assess the robustness of the results. The overall quality of the evidence is assessed using the Grading of Recommendations, Assessment, Development, and Evaluation (GRADE) framework.

**Data availability statement:** Data are shared on the Open Science Framework (Link: https://osf.io/pwamd/).

**Funding:** The author(s) received no specific funding for this work.

**Competing interests:** The authors have declared that no competing interests exist.

## Discussion

The planned systematic literature review will provide a synthesis of the current state of research on the comparative effectiveness of both treatment modalities. The objective is to furnish information for the delivery of effective patient care that also represents a cost-efficient solution for the healthcare system.

## Systematic review registration

This review protocol has been registered with the International Prospective Register of Systematic Reviews (PROSPERO) under the registration number CRD42023458888.

## Introduction

The prevalence of psychiatric crisis in childhood and adolescence has been poorly explored in scientific studies due to the lack of systematic data collection across European countries and the absence of a universally accepted definition [1–5]. The terms 'crisis' and 'emergency' are often used interchangeably in the literature and clinical practice. Establishing a clear distinction between the two can be challenging in practice, as the transitions can be frequently fluid [6,7]. A crisis in the psychosocial sense can arise from highly intense, extensive, and prolonged events (situational/ traumatic crisis) or life circumstances (developmental crisis) that overwhelm an individual's ability to cope [8]. If not adequately addressed or resolved, these crises may escalate into psychiatric emergencies [7]. Particularly in the paediatric population, psychiatric emergencies may occur in the context of a crisis, such as developmental crises associated with neurobiological maturation processes during puberty [1,6,9]. A crisis is not necessarily associated with acute danger. In contrast, psychiatric emergencies are defined by two key features: (1) an immediate threat to the patient's life or health due to self-harm or harm to others due to aggressive or dangerous behaviour, and (2) the need for immediate intervention in response to an acute condition. Psychiatric emergencies encompass a heterogeneous group of symptoms and disorders, that can manifest either as the initial onset or as an exacerbation of a psychiatric disease [6,7,10]. In this article, the term 'psychiatric crisis' refers to an acute situation requiring urgent treatment, characterized by an immediate threat or the perception of one. It may overlap with the concept of a 'psychiatric emergency' when a vital threat is present [11].

Experimental studies on psychiatric crises in children and adolescents are limited, partly due to the challenges of obtaining informed consent in acute situations. Existing studies identify the most common reasons for acute psychiatric presentations as nonsuicidal self-injurious behaviours (including anorexia nervosa), suicidal thoughts and actions, and behavioural disturbances with the potential to harm others. [12]. During the Coronavirus Disease 2019 (COVID-19) pandemic, a retrospective cohort study conducted in 63 hospitals in 25 countries showed a 50% increase in acute psychiatric presentations in children and adolescents worldwide [13].

The preferred treatment for acute psychiatric disorders is inpatient treatment [14–17]. This approach is rooted in the widely accepted belief that hospital treatment is the only safe and effective option for managing severe, often life-threatening psychiatric disorders [14,16]. Although inpatient treatment is sometimes unavoidable [17], it primarily focuses on symptom alleviation and involves the temporary separation of the affected individual from their family and social environment [15]. However, in contrast to psychiatric crises in adults, crises in children and adolescents should never be viewed as isolated incidents affecting only the individual. Instead, they must be understood as relational conflicts within the broader family and social context. It is therefore imperative that the severity of the crisis is always assessed in

relation to the familial environment [6,14]. Inpatient admission without sufficient involvement of the family system can lead to the development of institutional dependency. This occurs as the responsibility for crisis management is transferred to an external institution, preventing families from developing their own coping strategies for crisis situations. [14,16]. Additionally, the child or adolescent may return to an unchanged, dysfunctional, and stressful family environment, leading to a relapse and perpetuating the revolving door effect [17].

Given these challenges, it is imperative to explore accessible alternatives such as home-based interventions. Although the diversity of these interventions poses challenges in outcome comparison, they collectively emphasize family-based treatment [18]. In the field of adult psychiatry, the efficacy of acute home-based treatment has been investigated for clinical use, primarily based on the findings of two international meta-analyses. These studies demonstrated that home-based crisis intervention can reduce rehospitalizations by approximately 25% compared to inpatient care. Additionally, it has been shown to reduce symptom severity, improve patient and family satisfaction, and lessen the burden on families. There are also indications of enhanced cost-effectiveness, although the overall quality of evidence is low to moderate in this respect [19–21]. In general, there are few randomized controlled trials (RCTs) that evaluate the efficacy of psychiatric crisis interventions. This scarcity is mainly due to the ethical and practical challenges of enrolling patients in acute situations, especially children and adolescents. [22]. Nevertheless, the available studies, including a recent meta-analysis, show that home treatment leads to stable long-term outcomes. Home treatment has demonstrated comparable efficacy to inpatient care, especially in improving psychosocial functioning and reducing the severity of psychopathological symptoms [23–26]. Consequently, home treatment is increasingly viewed as a viable alternative. Studies also suggest that it may be more cost-effective, resulting in fewer inpatient bed days and lower overall costs [27, 28].

## Objectives

The existing research exhibits considerable heterogeneity, primarily due to variations in study populations, a diversity of included diagnoses, and different home treatment models implemented [23]. In this context, the research aims to systematically assess the effectiveness of home treatment models compared to inpatient treatment, with the focus on a more homogeneous group: children and adolescents experiencing psychiatric crises. This focus allows for a more detailed examination of the efficacy and applicability of home treatment approaches in this particularly vulnerable patient segment. The findings are important not only for the well-being of the people concerned, but also to reduce the burden on healthcare systems and social costs. The objective is (1) to provide an overview of the existing international home treatment models for children and adolescents in psychiatric crisis. (2) to evaluate the impact of these models on primary outcomes, including youth (social) functioning, psychopathology, and family functioning, in comparison to hospital admission. (3) to extract transition probabilities, such as the probabilities of rehospitalization, relapse, remission, and dropouts between different predefined health states as secondary outcomes. These secondary data will be used to develop a cost-utility analysis within a decision analytic model. The planned systematic literature review will focus exclusively on identifying the transition probabilities for the model and providing an overview of existing cost-effectiveness analyses. The standardization and adjustment of costs across OECD countries will be examined in a separate article.

## Definition and scope of home-based models

In this study protocol, the terms 'home treatment' and 'home-based' are used. These terms are often employed in a nonspecific manner, referring to a heterogeneous group of treatment

approaches that share a common outreach characteristic [29]. Therefore, it is important to delineate and describe their scope in this context.

The core concept of home treatment involves a multiprofessional treatment team providing care to acutely psychiatrically ill patients within their familiar environment. This care is delivered based on an agreed-upon treatment plan and ensures round-the-clock availability of the team. The duration of these interventions is intended to be comparable to, or shorter than, that of a hospital stay [30–32]. In the literature, home treatment is not clearly distinguished from other community care services. Concepts such as assertive community treatment (ACT), case management (CM), or wraparound services are often interpreted as forms of home treatment [33,34]. Unlike home treatment, which is conceptualized as a true alternative to inpatient treatment, these approaches serve as long-term complementary services. They are aimed at stabilizing and rehabilitating individuals with chronic mental health conditions within the community [21,31]. In this regard, intensive community care models such as the Supported Discharge Service (SDS) from England [27,35] and the Hot-BITs-treatment (home treatment brings inpatient-treatment outside) from Germany [25,26,28] are noteworthy. Both concepts combine inpatient and outpatient elements by incorporating a shortened inpatient stay for crisis stabilization, followed by continued treatment in the home setting. In the following sections, these will be referred to as 'mixed interventions'.

A standalone form of home treatment originating in the USA is Multisystemic Therapy (MST), which has proven effective in reducing antisocial behavior and institutional placements among chronic and violent juvenile offenders [36]. Henggeler and his team modified MST to treat psychiatric and suicidal crises in children and adolescents in the home setting [37]. Another form is represented by Crisis Resolution and Home Treatment Teams (CRHTTs). In the literature, these teams are referred to by various terms, including 'crisis resolution team', 'crisis assessment and treatment team', 'intensive home treatment team', and 'mobile crisis intervention' [38]. The concept originated in the USA and Australia. It was later adopted on a national level in England as part of the NHS plan, resulting in the establishment of 335 CRHTTs by 2004. It also serves as a model for ward-equivalent treatment (Stationsäquivalente Behandlung, StäB) in Germany. CRHTTs offer short-term, acute outreach treatment aimed primarily at preventing inpatient admissions and pediatric emergency visits in somatic hospitals. These are often inadequately equipped to treat adolescents with behavioral problems. Depending on the specific design of the CRHTTs, self-harm or harm to others is often considered a contraindication for home treatment [38–41].

## How the intervention might work

To illustrate the interactions as causal relationships between the key elements analyzed in this review, a preliminary logic model has been created (Fig 1). The model includes both mixed interventions, which involve home-based interventions following a shortened inpatient stay. It also includes stand-alone home-based treatments, where the treatment takes place in the home environment from initiation to completion. However, the model exclusively visualizes the period from home treatment onwards.

In both cases, treatment initiation depends on several factors: the therapist's assessment of whether home treatment is appropriate, the availability of a trained home treatment team, appropriate infrastructure, and parental consent or a family preference for conducting treatment in the home environment. Additionally, the home environment must be suitable for conducting both individual and family sessions [15,18].

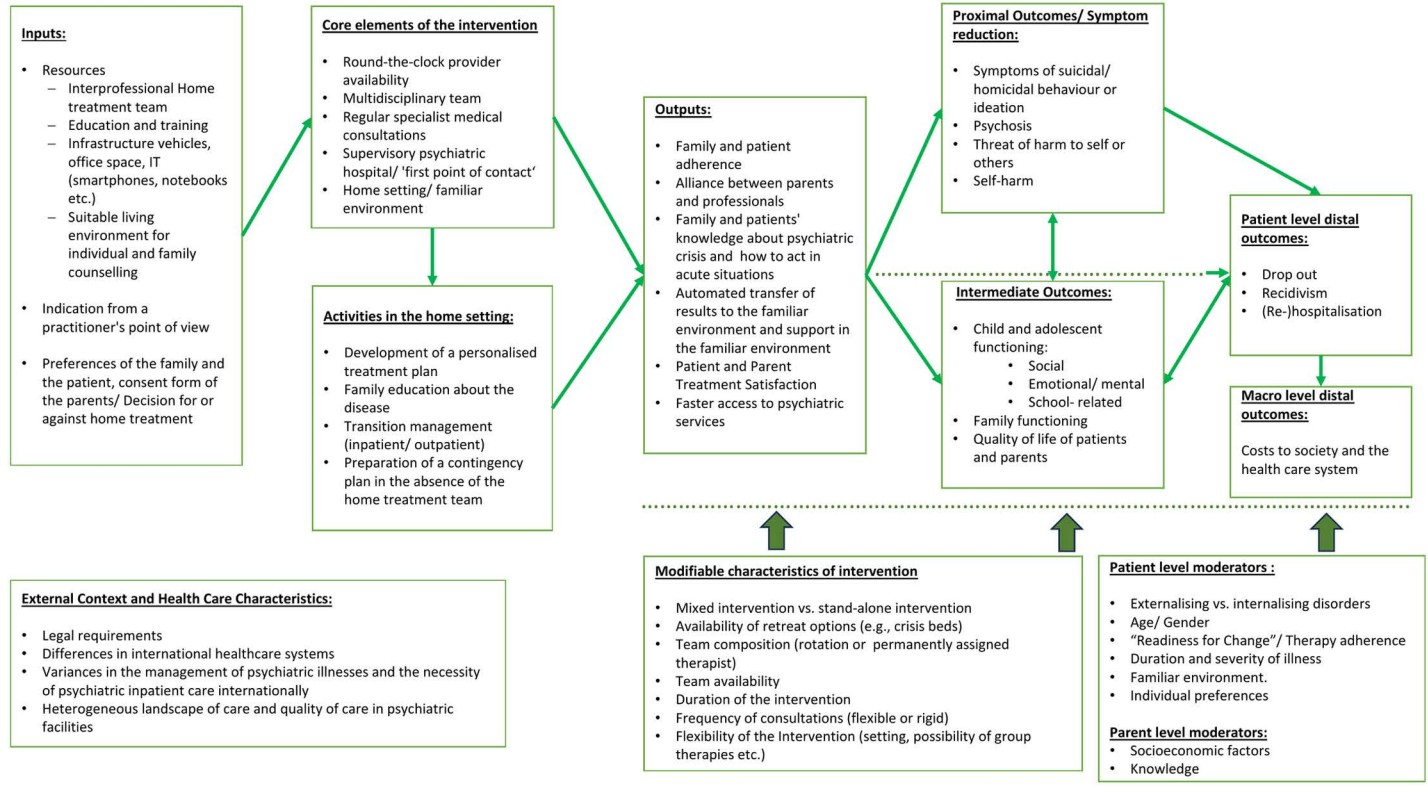

**Fig 1. Logical pathway model for home-based interventions to manage psychiatric crises in children and adolescents.**

The treatment models outlined in the logic model include core elements. These include a diagnosis that necessitates inpatient treatment with 24-hour access to a multidisciplinary team and regular specialist consultations. A central coordinating entity, usually a psychiatric entity, serves as the primary point of contact. It organizes the home treatment team and provides essential resources such as vehicles, training, and IT equipment.

Home treatment modalities are characterized by their family-centered approach, which requires greater commitment and responsibility for relatives than other treatment settings [15]. Through effective communication and collaboration between patients, clinicians, and family members, parents may gain a better understanding of the development and management of psychiatric crises. This family education may allow parents to develop skills for recognizing acute situations and intervening to prevent or de-escalate the recurrence of such crises [1]. The collaborative development of therapy plans and the creation of contingency plans for situations in which the home treatment team is unavailable contribute to increased parental involvement and confidence in managing psychiatric crises. This increased confidence strengthens the trust of the children in their caregivers and support system. Overall, home treatment represents an opportunity to foster a trusting relationship between patients, parents, and caregivers, potentially cultivating a collaborative therapeutic alliance [15]. This can be facilitated by the close supervision provided, potentially resulting in improved therapy adherence for children and adolescents.

The identification of symptoms and potential triggers in the home environment may also reduce transfer problems during a transition from the hospital to familiar surroundings and therefore may reduce the risk of rehospitalization and relapse [15]. Rehospitalization often

occurs due to inadequate maintenance of therapeutic progress. This is especially true when there is a sudden transition to the home environment after discharge from psychiatric facilities, particularly without established outpatient follow-up care [42]. Moreover, remaining in the familiar environment, rather than in a psychiatric facility where interactions with other patients could exacerbate maladaptive behavior, can also positively impact symptom reduction [43].

Assuming that children or adolescents prefer to be in their home environment, their satisfaction with treatment may increase as they are not removed from their familiar surroundings. They can potentially continue to interact with friends and maintain their normal lives, provided there is no longer a risk of harm to themselves or others. This may also facilitate access to the support system, as there is no fear of being stigmatized or isolated in an already vulnerable situation.

Considering the effects mentioned above, home-based interventions may have the potential to positively impact family functioning, as well as the individual functioning of children or adolescents, thereby improving the quality of life for both patients and parents. Coupled with increased treatment adherence and support in the home environment, these interventions could lead to symptom improvement. Furthermore, improved family and social functioning, combined with a reduction in symptoms, may result in fewer relapses and rehospitalizations. This could, in turn, lead to lower costs for society and the healthcare system on the macro level.

**External context/ modifiable characteristics of intervention/ child or adolescent level and parent-level moderators.** The depiction outlined above needs to be supplemented by acknowledging that the implementation of home-based interventions depends on on external factors and the context in which they are conducted. This includes, the diverse international legal frameworks governing home treatment, the variety of healthcare systems with potential disparities in the management of psychiatric disorders, and the heterogeneous landscape and quality of care within psychiatric facilities.

In addition to the external context, it is also crucial to identify important modifiable characteristics, as they can have a direct impact on the output and outcome parameters mentioned. Therefore, it is crucial to consider options as the temporary removal from the home environment through the availability of crisis beds or the possibility of full transfer to a psychiatric ward. This becomes especially significant if a patient requires transfer to a secure setting due to severe symptoms or when temporary removal from their family environment is necessary to facilitate crisis management in a neutral environment [44]. Modifiable characteristics include various aspects such as team composition, which can range from rotating therapists to a dedicated reference therapist who provides ongoing support to the family. In addition, the duration of intervention can be short-term for immediate de-escalation or long-term for sustained support. Team availability and visit frequency can also vary, from rigid schedules to flexible arrangements. Flexibility in the intervention is also crucial, allowing for activities such as school visits or meetings in neutral locations. Additionally, participation in group therapies and activities within the inpatient setting may take a variety of forms. For instance, it facilitates the analysis of peer interactions and provides patients with the opportunity to share experiences with others facing similar challenges, thereby enhancing their treatment experience and offering essential social support.

Patient-specific factors, such as the type of disorder, age, and gender [15,18], as well as the general attitude towards the treatment modality and readiness to change, can also influence treatment outcomes. The model assumes that patients generally prefer to receive treatment in their home environment, although this may not always be the case. In acute

situations, patients may have limited involvement in the decision-making process regarding home treatment, which can negatively impact both satisfaction and treatment adherence. Another factor that must be considered a confounder is the baseline attitude of parents, as well as their knowledge of the disorder and socioeconomic status. It is reasonable to suggest that families with a supportive environment from the start may be more inclined towards a treatment approach like home treatment. This approach involves greater family involvement and commitment and often requires a more substantial understanding of crisis management. Additionally, socioeconomic status can influence awareness of psychiatric illness and potentially provide better home conditions for treatment [45]. However, these factors also present limitations in terms of examining treatment success, as they may be challenging to quantify or prove definitively.

## Methods

### Reporting standards and registration

The Preferred Reporting Items for Systematic Reviews and Meta-Analyses (PRISMA) guidelines will be used as the methodological framework for conducting the systematic review (S1 Table) [46]. To ensure transparency and help prevent duplication of efforts, the protocol for this study is registered in the International Prospective Register of Systematic Reviews (PROSPERO) under the registration number CRD42023458888.

### Eligibility criteria

**Study design.** For the systematic literature review, quantitative primary studies with specified study designs should be considered:

- Randomized trials

- Cluster randomized trials

- Nonrandomized trials

- Controlled before-after studies

- Controlled interrupted time series studies

- Controlled cohort studies [47].

Overall qualitative studies, noncontrolled studies, and cross-sectional studies will be excluded [48].

**Population.** The systematic review includes children and adolescents under the age of 18. It focuses exclusively those living at home with stable, permanent caregivers (e.g., parents, relatives, or foster families). Individuals residing in institutional settings (e.g., homeless shelters, group homes) will be excluded. There will be no restrictions regarding socioeconomic status, ethnicity, or cultural background. Studies with mixed populations (e.g., adolescents and adults) will only be included if the subgroups are clearly defined and if it is possible to extract data from the whole population. Any study where extraction is not feasible will be excluded.

The systematic review includes children and adolescents suffering from an acute mental illness or symptoms that typically necessitate acute psychiatric hospitalization due to their severity. These conditions may present as a first manifestation or as an exacerbation of an existing disorder. Comorbid conditions will be included as clear distinctions are often challenging to make in crisis situations. For instance, externalizing conduct disorders and internalizing emotional disorders frequently overlap and co-occur [49].

**Included psychiatric disorders and symptoms in the context of psychiatric crisis (according to ICD-10-CM):**

◦ Suicide attempt (T14.91), suicidal ideation (R45.851)

◦ Mood (affective) disorders/ depression (F30-F39)

◦ Anxiety (F40-F41)

◦ Reaction to severe stress and adjustment disorders (F43) (acute stress reaction, PTSD, adjustment disorders)

◦ Psychomotor agitation/ violent or disruptive behavior (stand-alone (R45.-) or as a symptom of an underlying disorder, such as conduct disorder (F91.2))

◦ Nonsuicidal self- harm (stand-alone (R45.88) or as a symptom of an underlying disorder, such as borderline personality disorder (F60.3))

◦ Stupor (R40.1) and Catatonia (F06.1)

◦ Mania (stand-alone (F30) or as a symptom of an underlying disorder, such as bipolar disorder (F31.-))

◦ Acute psychosis (stand-alone (F23) or as a symptom of an underlying disorder, such as schizophrenia (F20-F29) [49–51].

**Excluded psychiatric disorders and symptoms in the context of psychiatric crisis (according to ICD-10-CM):**
The exclusion criteria encompass diagnoses with a predominantly chronic nature, such as encopresis, enuresis, tic disorders, and chronic courses of other psychiatric conditions. Mixed populations, which include both acute and chronic cases, will only be considered if they can be clearly identified and distinguished separately within the studies. Patients who require restrictive measures, such as isolation or restraint, due to their medical condition will be excluded. Additionally, patients with severe somatic symptoms will also be excluded. This involves:

◦ Intoxication States or Withdrawal (F10-F19)

◦ Anorexia nervosa or bulimia nervosa with imminent somatic decompensation (F50)

To ensure a homogeneous base population, individuals with the following conditions will also be excluded:

◦ Intellectual disabilities (F70-F79) in conjunction with psychiatric comorbidities

◦ Developmental disorders (F80-89), such as autism.

**Experimental intervention.** Only home- based interventions that are a viable alternative to hospitalization will be included. These may be interventions that take place in the home environment from admission to discharge, or a combined treatment consisting of a shortened inpatient stay followed by home treatment (mixed interventions). In either case, the following criteria must be met with respect to the home care setting for both models:

◦ Treatment must be provided primarily in the home environment, but may be provided in other natural settings, such as schools, when appropriate. Treatment must be provided outside of a hospital setting, with a primary focus on evaluating the effectiveness of home treatment in the context of intensive parent-child interactions.

○ Treatment must be provided by a multidisciplinary team.

○ Specialist medical consultations are mandatory.

○ The multidisciplinary team must be available on a 24-hour basis.

Treatment modalities meeting the following criteria will be excluded:

○ Treatment provided in supervised group homes, youth facilities, or homeless shelters.

○ Interventions carried out exclusively by a single professional group such as social workers or nurses.

Overall, a comprehensive treatment approach from admission to discharge planning is needed. Short-term support measures provided by crisis intervention centers, counselling services, helplines, etc., will be excluded. In many studies, there is an insufficient distinction between home treatment and community treatment. For this analysis, community treatment will be only included if it specifically addresses acute cases. Services intended to stabilize chronically ill patients in the community, will be excluded from this study. Studies that consider treatment at home as a solely preventive measure will be also excluded.

**Control intervention.** The comparative intervention is inpatient treatment in a psychiatry as the current standard of care, which includes 24/7 supervision, to determine relative effects. Studies that use emergency departments or somatic hospital wards as control groups will be excluded from the investigation.

## Information sources

The bibliographic databases Medline, Cochrane Library, Embase, and PsycInfo will be searched first. In addition to the academic literature, grey literature sources and hand searches will be explored to ensure a comprehensive search and reduce the risk of publication bias [52]. The first 50 hits in electronic grey literature databases, including Google Scholar and the System for Information on Grey Literature in Europe (OpenGrey), will be searched to find grey literature such as reports, preprints, and conference abstracts. To identify ongoing or unpublished trials, the trial registries ClinicalTrials.gov and the World Health Organization (WHO) International Clinical Trials Registry Platform (ICTRP) will be searched. If necessary, relevant authors and organizations will be contacted to obtain information about unpublished or ongoing trials. In cases where supplementary material is needed, the authors of the original trials will be contacted.

## Search strategy

A highly sensitive search strategy for Medline was developed and tested by the author team (S2 Table). This strategy will be adapted to meet the specific requirements of other selected bibliographic databases. The search terms consist of a combination of Medical Subject Heading (MeSH)- terms and predefined keywords categorized by population, intervention, and control intervention. To maximize sensitivity, the search strategy does not include keywords related to outcome parameters. The search for the same keywords is also conducted in the mentioned gray literature databases that do not have syntax/search operators. There are no restrictions on the publication period. However, only studies in German and English are included and identified using the search filter function. No additional search filters will be used. After identifying relevant full texts, reference harvesting and backwards (and forward) snowball searches are conducted for the included studies to enable the most comprehensive search possible.

## Study records

**Data management and study selection.** The search results are stored using the web application Rayyan, which is also used for deduplication. Relevant studies are selected stepwise using the predefined inclusion and exclusion criteria. First, titles and abstracts are screened in Rayyan, followed by full-text screening. The screening and selection process involves at least two independent reviewers for each step, with any conflicts resolved by a third reviewer. The results of the inclusion and exclusion process for all studies are presented in a PRISMA flowchart that illustrates the entire screening and selection process [46]. The files of the included studies, the data collection forms, and the reference lists are available to all the authors via internet-based exchange options (e.g., Rayyan or email).

**Data collection and extraction.** The data from the included full texts will be manually extracted. Information requiring subjective interpretation, as well as data crucial for interpreting the results (outcome data), will be extracted independently by at least two reviewers to minimize errors and reduce the potential introduction of bias [53]. Relevant data will be extracted in a standardized and systematic way, using previously developed structured tables. In the initial step, the study parameters and outcomes, including observation periods and measurement methods of the included studies, will be presented descriptively [54].

Studies indicate that between 31% and 38% of discharged children and adolescents are readmitted to the psychiatry clinics within 12 months, with most of these rehospitalizations occurring within 90 days of discharge [42,55–59]. Against this background, outcomes will be assessed at the following intervals: Less than one month, to capture the immediate post-discharge period during which most readmissions occur. Three to six months, to evaluate the stability of treatment effects mid-term. And six to twelve months, to assess long-term outcomes and provide a comprehensive overview of the durability of psychiatric interventions over time.

## Outcomes and data items

The focus will be on studies reporting at least one of the following primary outcomes, which can be reported by children or adolescents, parents, or caregivers:

- Youth functioning: Assessed using instruments such as the Child Behavior Checklist (CBCL)

    ◦ Social functioning (e.g., maintaining friendships)

    ◦ Mental well-being (e.g., self-esteem, positive expectations for the future.)

    ◦ School-related functioning (e.g., school attendance)

- Symptom severity: Measured using instruments such as the Global Severity Index (GSI), Health of the Nation Outcome Scales for Children and Adolescents (HoNOSCA) or the Brief Symptom Inventory (BSI). This also includes suicidal ideation and attempted suicide, which are measured using instruments such as the Youth Risk Behavior Survey (YRBS) to investigate suicidal thoughts and suicide attempts among patients.

- Family functioning: Evaluated using tools such as the Family Adaptability and Cohesion Evaluation Scale (FACES).

- Quality of life among children/ parents: assessed using units such as the EQ-5D, or SF-6D.

The systematic review will serve as the foundation for subsequent health-economic evaluation using a decision-analytic modelling approach. The following outcomes are intended to inform model input parameters such as transition probabilities:

- Admission rate to inpatient care during home treatment

- Likelihood of treatment discontinuation/drop-out

- Length of inpatient stay

- Number of suicides in terms of objective number of deaths/hospitalizations for suicide attempts (mortality)

- Relapse rate

- (Re)hospitalization rate

- Remission rate

- Cost- Effectiveness

## Risk of bias

The RoB2 tool will be used for (cluster) randomized studies. Nonrandomized studies will be assessed using the ROBINS-I tool. Nonrandomized intervention studies, such as controlled cohort studies, interrupted time series studies, and controlled before-after studies, will also undergo evaluation using the ROBINS-I [60,61]. Given the use of the ROBINS-I tool and the potential impact of attrition bias, whether an intention-to-treat (ITT) methodology was used will be recorded. In addition, in sensitivity analysis the impact of adherence to the per-protocol approach versus nonadherence will be assessed [60].

The risk of bias will be assessed by at least two independent authors. In cases of disagreement, a third author will resolve conflicts and reach a consensus. The risk tables will present assessments of bias risk as high, low, or with some concerns, with sources for the assessments provided [61].

## Data synthesis

Depending on the measurement method, dichotomous, continuous, and ordinal data will be extracted. For dichotomous data, the primary synthesis method is the calculation of the risk ratio (RR). In instances where RR cannot be computed, the odds ratio (OR) or risk difference will be employed as alternatives. Continuous data will be presented as the mean difference (MD) whenever feasible [62,63]. The data extracted for ordinal outcomes will either be dichotomized or treated as continuous outcomes. Furthermore, if a substantial number of ordinal outcomes are extracted, they may also be analyzed as ordinal data [63].

To determine the feasibility of quantitative synthesis through meta-analysis, the clinical heterogeneity of therapeutic effects will be assessed using Cochrane's Q chi-square ($\chi^2$) test and Higgins's and Thompson's I-squared ($I^2$) tests. These metrics will be presented in a forest plot diagram. Due to the expected heterogeneity across studies during the preliminary review, a random-effects meta-regression for data synthesis will be used, following the DerSimonian and Laird method. This analysis will be performed using R-Studio (version R 4.3.2) [64,65]. In cases of substantial heterogeneity, defined by an $I^2 > 50\%$, indicating significant diversity among individual studies, a meta-analysis will not be conducted [65]. Instead, study results demonstrating insufficient homogeneity will be presented narratively [66].

## Subgroup analyses and sensitivity analyses

In cases of statistical heterogeneity, subgroup analyses and sensitivity analyses will be conducted to explore potential causes of heterogeneity [64]. If possible, data will be stratified

based on factors likely contributing to heterogeneity: clinical, demographic, geographical characteristics, variations in home-based treatment modalities, or differences in the measurement instruments used for predefined outcomes. In sensitivity analyses, individual studies that could exert a strong influence on the outcome or those exhibiting high risk of bias will be excluded from the analysis [67]. Whether subgroup or sensitivity analyses will be conducted depends on the number of studies identified. These analyses will be performed if at least three studies evaluate the effects of comparable interventions on the same outcome using a similar control group. In the case of a narrative literature review, differences will be identified during the structured data extraction process and presented in tabular format. As part of the data synthesis, the results will then be discussed narratively within and between subgroups.

Heterogeneity among the included studies is to be expected, among other factors, due to the variety of included symptoms and the differing severity of psychiatric crises. Socioeconomic status differences may also influence the reported outcome parameters. Additionally, variations in international healthcare systems and the diversity in treatment model designs further contribute to these differences.

### Meta-bias(es)

To examine the potential impact of publication bias within the scope of the meta-analysis, funnel plots for the primary outcomes will be constructed and assessed using the statistical Egger test. The Egger's test requires a minimum of 20 studies to demonstrate adequate power [68,69]. Therefore, this analysis will be conducted only if there are at least 20 studies that assess comparable interventions with the same outcomes and similar control groups.

### Confidence in cumulative evidence

The Grading of Recommendations Assessment, Development and Evaluation (GRADE) tables will be employed to provide a comprehensive assessment of the quality of evidence and strength of recommendations for the primary endpoints. To ensure accuracy and consistency, at least two authors will be involved in this assessment, and any discrepancies will be resolved through discussion with a third author [70].

## Discussion

A substantial proportion of children and adolescents who require psychiatric assistance receive no help at all or only obtain it during an acute stage of their illness. [71]. Delayed help-seeking is often caused by factors such as long waiting times and concerns about potential stigma associated with admission to a mental health facility [72]. If help is sought in acute cases, this is predominantly done in an inpatient setting. However, in these cases there is a high rate of rehospitalization, e.g., in suicidal patients one year after discharge with a relapse rate of 30-43% [73,74], which leads to further individual suffering and a continuing financial burden on the health and social system [18]. The systematic review will focus on synthesizing the best available evidence on home treatment for young patients in psychiatric crisis. The study will draw conclusions and provide evidence-based recommendations on the effectiveness and cost-effectiveness of home-based treatment approaches compared with traditional inpatient treatment. The knowledge gained from this study can help healthcare decision makers and clinical guideline developers formulate improved recommendations to ensure the best possible care for young patients. In accordance with PRISMA guidelines, a highly sensitive search strategy will be employed to ensure comprehensive identification of relevant studies. The search will be conducted across four major databases—Medline, Embase, Cochrane, and PsycInfo—with the focus on children and adolescents in psychiatric crises. A challenge in

this research project is the heterogeneity of home treatment models, which vary in terms of design, definition, and international frameworks. Therefore, during the preliminary database search, special attention was given to adhering to the strictly defined inclusion and exclusion criteria.

To reduce the heterogeneity of the study population, only OECD countries will be included. However, it is important to note that even these countries exhibit differences in their healthcare system structures, which may potentially influence the outcome parameters. For instance, international differences in inpatient bed availability may influence hospitalization practices. Countries with higher bed capacities, such as Germany—with an average stay of 33.7 days and the highest bed availability in Europe—are more inclined to admit patients for inpatient care [75]. In contrast, countries like the USA, with an average stay of 5–8 days, primarily admit only severe cases and focus on rapid stabilization in crises, often within 24 hours [76]. Differences in access to psychiatric healthcare services are also evident. In market-oriented systems like the USA, financial barriers are higher compared to contribution-funded systems such as Germany's or primarily tax-funded systems like the NHS in England [77,78]. Differences in billing frameworks and implementation options for home treatment between countries can also impact its effectiveness. For instance, while other countries can implement flexible, individualized care plans, Germany follows stricter regulations for its home treatment model (StäB = inpatient-equivalent treatment). According to the Operation and Procedure Classification System (OPS) standards, at least one in-person patient contact per day is required for correct billing, limiting flexibility in care delivery [79]. Another challenge is the specific population of children and adolescents in psychiatric crises. Due to ethical considerations, it is likely that there are few studies with control groups. Given the mentioned heterogeneity, careful consideration will be required to determine whether a meta-analysis is feasible or whether alternative methods of synthesis may be more appropriate if the heterogeneity proves to be too high (when $I^2 > 50\%$). The search results will also be updated regularly to ensure the results remain reliable.

## Supporting information

**S1 Table. PRISMA-P checklist.**
(PDF)

**S2 Table. Search strategy for Medline database.**
(PDF)

## Author contributions

**Conceptualization:** Karolina Foremnik, Gaby Sroczynski, Jan Stratil, Anja Neumann, Barbara Buchberger.

**Data curation:** Karolina Foremnik.

**Formal analysis:** Karolina Foremnik.

**Funding acquisition:** Karolina Foremnik.

**Investigation:** Karolina Foremnik.

**Methodology:** Karolina Foremnik, Gaby Sroczynski, Jan Stratil, Anja Neumann, Barbara Buchberger.

**Project administration:** Karolina Foremnik, Barbara Buchberger.

**Resources:** Karolina Foremnik.

**Supervision:** Barbara Buchberger.

**Validation:** Barbara Buchberger.

**Visualization:** Karolina Foremnik.

**Writing – original draft:** Karolina Foremnik.

**Writing – review & editing:** Gaby Sroczynski, Jan Stratil, Anja Neumann, Barbara Buchberger.

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
