## [Decision Letter · Decision Letter 0]

19 Nov 2024

PONE-D-24-45550

Efficacy of home-based and inpatient treatment for children and adolescents in psychiatric crisis: A systematic review protocol

PLOS ONE

Dear Dr. Foremnik,

Thank you for submitting your manuscript to PLOS ONE. After careful consideration, we feel that it has merit but does not fully meet PLOS ONE’s publication criteria as it currently stands. Therefore, we invite you to submit a revised version of the manuscript that addresses the points raised during the review process.

**Comments from the editorial office** : Upon internal evaluation of the reviews provided, we kindly request you to disregard the reviewer report provided by Reviewer 1. No amendments are required in response to Reviewer 1’s comments. 

We look forward to receiving your revised manuscript.

Kind regards,

Cho-Hao Howard Lee, M.D.

Academic Editor

PLOS ONE

Reviewers' comments:

Reviewer's Responses to Questions

**Comments to the Author**

1. Does the manuscript provide a valid rationale for the proposed study, with clearly identified and justified research questions?

Reviewer #1: Partly

Reviewer #2: Yes

Reviewer #3: Yes

Reviewer #4: Partly

2. Is the protocol technically sound and planned in a manner that will lead to a meaningful outcome and allow testing the stated hypotheses?

Reviewer #1: Partly

Reviewer #2: Yes

Reviewer #3: Yes

Reviewer #4: Yes

3. Is the methodology feasible and described in sufficient detail to allow the work to be replicable?

Reviewer #1: No

Reviewer #2: Yes

Reviewer #3: Yes

Reviewer #4: No

4. Have the authors described where all data underlying the findings will be made available when the study is complete?

Reviewer #1: Yes

Reviewer #2: Yes

Reviewer #3: Yes

Reviewer #4: Yes

5. Is the manuscript presented in an intelligible fashion and written in standard English?

Reviewer #1: Yes

Reviewer #2: Yes

Reviewer #3: Yes

Reviewer #4: Yes

6. Review Comments to the Author

You may also provide optional suggestions and comments to authors that they might find helpful in planning their study.

Reviewer #1: Thank you for the opportunity to review this protocol. It’s clear that you’ve put a lot of thought into tackling a critical area of child and adolescent mental health, especially given the rise in psychiatric crises among youth. Your approach to systematically comparing home-based and inpatient care is ambitious and has the potential to provide valuable insights for both practitioners and policymakers. That said, there are some areas that, with additional refinement, could strengthen the overall impact and clarity of the manuscript.

Merits

1. Timely and Meaningful Topic: This study addresses a highly relevant topic, particularly post-COVID-19, as many healthcare systems are evaluating alternatives to traditional inpatient psychiatric care. The comparative analysis of home-based and inpatient treatments can provide critical insights for improving youth mental health services, especially in reducing stigma and enhancing family involvement.

2. Structured and Rigorous Methodology: Your adherence to PRISMA guidelines and the planned use of established assessment tools like RoB2, ROBINS-I, and GRADE demonstrates a thorough and organized approach. This rigor is essential in producing findings that stakeholders can trust and act upon.

3. Holistic Focus on Key Outcomes: The range of primary outcomes (like family functioning and psychopathology) and secondary outcomes (such as hospitalization rates) is impressive. This thoughtful outcome selection ensures a comprehensive look at both the direct and broader effects of these treatment models on young people and their families.

Suggestions for Major Revision

1. Clarify Study Selection Criteria: The inclusion and exclusion criteria are a bit broad. Narrowing down these criteria, or offering more specifics, would help the reader understand exactly which studies you aim to include. This will also make it easier for future readers and researchers to interpret the findings within a well-defined scope.

2. Strengthen the Approach to Handling Heterogeneity: Given the diversity in intervention types and study designs, there’s likely to be substantial heterogeneity. It would be helpful to elaborate on how you plan to handle this in your synthesis. Specifically, describing when you might use meta-analysis versus narrative synthesis would clarify your approach and strengthen your methods section.

3. Acknowledge Potential Challenges: Considering the complexities involved in reviewing psychiatric interventions, addressing potential limitations upfront—such as the limited number of RCTs in this field or inconsistencies in intervention definitions—would provide a more balanced picture. This could help readers understand the potential scope and constraints of your review more clearly.

4. Expand on Subgroup Analysis Plans: You mentioned possible subgroup analyses by factors like age and intervention type. Specifying exactly how you intend to conduct these analyses would provide additional clarity, particularly for clinicians and researchers interested in how different demographics or intervention models impact outcomes.

5. Highlight Practical Implications for Healthcare: Since this review could inform clinical and policy decisions, adding a few thoughts on the practical implications could enhance its relevance. Outlining how the findings might translate to on-the-ground changes for clinicians or system planners could make this protocol even more compelling.

Recommendation: Major Revision

In sum, this protocol addresses an important topic with real-world relevance, and I believe it has significant potential. With some refinements to the scope and methodology, it will be even more robust and ready for publication. I look forward to seeing the revised version, as I think your findings will make a meaningful contribution to the field. Thank you again for sharing your work—it’s exciting to see research that could shape future directions in youth mental health care.

Reviewer #2: This paper presents a systematic review protocol aimed at comparing the efficacy of home-based versus inpatient treatment for children and adolescents experiencing psychiatric crises.

The protocol outlines a comprehensive methodology to evaluate treatment outcomes across multiple domains, including psychopathology, family functioning, and social functioning. It proposes to analyze both primary clinical outcomes and secondary parameters such as readmission rates and cost-effectiveness. The methodology includes a systematic search of major databases, rigorous quality assessment using standardized tools (RoB2 and ROBINS-I), and meta-analysis where appropriate.

1. The protocol's criteria for "psychiatric crisis" The protocol lacks precise clinical parameters for defining "psychiatric crisis," which could lead to inconsistencies in study selection and comparability. It is encourage that have a brief of explain about psychiatric crisis in the introduction part.

2. Figure Clarity: The figures in the paper are unclear, making it difficult to discern specific details. It is recommended to upload high-resolution images.

3. Line 22 - Author Attribution: Only one author is marked, which may lead to misunderstandings regarding contributions.

Addressing these limitations will enhance the scientific rigor and practical applicability of the study's findings.

Reviewer #3: This systematic review protocol aims to evaluate and compare the effectiveness and cost-effectiveness of home-based and inpatient treatments for children and adolescents in psychiatric crises. The authors propose a rigorous methodological approach, adhering to the PRISMA guidelines and incorporating a structured risk assessment using the Cochrane RoB2 and ROBINS-I tools. The study is valuable as it addresses a significant gap in understanding alternative psychiatric care for young individuals, offering potential insights into less restrictive, more family-centered treatment options.

Below are specific points to consider for enhancing the manuscript:

1. While the manuscript specifies the inclusion of children and adolescents under 18 from OECD countries, further clarity on the demographic and clinical characteristics (e.g., psychiatric diagnoses) that will be included would enhance reproducibility and relevance.

2. Clarifying the operational definitions for “home-based” interventions could strengthen the manuscript, as these can vary significantly across settings and countries.

3. Including a brief justification for the chosen follow-up intervals (e.g., 1, 3-6, and 6-12 months) and their relevance to long-term outcomes would enhance readers' understanding of these selections.

Reviewer #4: The manuscript presents a systematic review protocol to evaluate the effectiveness and cost-effectiveness of home-based treatment models for managing psychiatric crises in children and adolescents. This study addresses a significant gap and could provide valuable insights for healthcare providers and policymakers.

Major Comments:

1. Language could be streamlined for better readability. Consider rephrasing to avoid complex, multi-clause sentences that may hinder reader comprehension.

2. The scope of inclusion and exclusion criteria regarding mixed models and family-based intervention specifics could benefit from further clarification. For example, specifying clear distinctions between what constitutes "mixed interventions" versus standalone models would help ensure replicability of the protocol.

3. The decision to limit the population to OECD countries to reduce heterogeneity is commendable, but further justification would be more beneficial as some OECD countries may have different baseline healthcare structures and outcomes which could impact results.

4. Given the diversity of interventions included, the review might also benefit from incorporating additional bias assessments relevant to intervention studies. For example, the potential for performance bias, particularly in non-randomized studies where blinding may be challenging.

5. A brief overview of the decision-analytic framework and how cost data (particularly for various health systems within OECD countries) will be standardized or adjusted would be beneficial for clarity.

Minor Comments:

1. The manuscript occasionally uses varying terminology (e.g., “rehospitalization” vs. “readmission”). Consistent use of terminology throughout the text will improve clarity.

2. Lines 425–427: the choice of 10 studies as a minimum threshold could be briefly justified or referenced to support this decision.

In summary, this systematic review protocol addresses an important research gap by comparing home-based and inpatient psychiatric care models for children and adolescents in crisis. With revisions, this study protocol could make a significant contribution to pediatric mental health research.

7. PLOS authors have the option to publish the peer review history of their article (what does this mean? ). If published, this will include your full peer review and any attached files.

**Do you want your identity to be public for this peer review?** For information about this choice, including consent withdrawal, please see our Privacy Policy .

Reviewer #1: **Yes: ** Xiaoyi Zhang, M.D.

Reviewer #2: No

Reviewer #3: No

Reviewer #4: **Yes: ** Yuhang Liu

---

## [Author Response · Author response to Decision Letter 1]

21 Dec 2024

Dear Editorial Team,

we would like to express our gratitude for the constructive comments and valuable suggestions regarding the manuscript. We have carefully reviewed the remarks and implemented the necessary changes to enhance the quality of the study protocol. In the following sections, we will address each of your comments in turn.

Reviewer #1: Thank you for the opportunity to review this protocol. It’s clear that you’ve put a lot of thought into tackling a critical area of child and adolescent mental health, especially given the rise in psychiatric crises among youth. Your approach to systematically comparing home-based and inpatient care is ambitious and has the potential to provide valuable insights for both practitioners and policymakers. That said, there are some areas that, with additional refinement, could strengthen the overall impact and clarity of the manuscript.

Merits

1. Timely and Meaningful Topic: This study addresses a highly relevant topic, particularly post-COVID-19, as many healthcare systems are evaluating alternatives to traditional inpatient psychiatric care. The comparative analysis of home-based and inpatient treatments can provide critical insights for improving youth mental health services, especially in reducing stigma and enhancing family involvement.

2. Structured and Rigorous Methodology: Your adherence to PRISMA guidelines and the planned use of established assessment tools like RoB2, ROBINS-I, and GRADE demonstrates a thorough and organized approach. This rigor is essential in producing findings that stakeholders can trust and act upon.

3. Holistic Focus on Key Outcomes: The range of primary outcomes (like family functioning and psychopathology) and secondary outcomes (such as hospitalization rates) is impressive. This thoughtful outcome selection ensures a comprehensive look at both the direct and broader effects of these treatment models on young people and their families.

Suggestions for Major Revision

1. Clarify Study Selection Criteria: The inclusion and exclusion criteria are a bit broad. Narrowing down these criteria, or offering more specifics, would help the reader understand exactly which studies you aim to include. This will also make it easier for future readers and researchers to interpret the findings within a well-defined scope.

2. Strengthen the Approach to Handling Heterogeneity: Given the diversity in intervention types and study designs, there’s likely to be substantial heterogeneity. It would be helpful to elaborate on how you plan to handle this in your synthesis. Specifically, describing when you might use meta-analysis versus narrative synthesis would clarify your approach and strengthen your methods section.

3. Acknowledge Potential Challenges: Considering the complexities involved in reviewing psychiatric interventions, addressing potential limitations upfront—such as the limited number of RCTs in this field or inconsistencies in intervention definitions—would provide a more balanced picture. This could help readers understand the potential scope and constraints of your review more clearly.

4. Expand on Subgroup Analysis Plans: You mentioned possible subgroup analyses by factors like age and intervention type. Specifying exactly how you intend to conduct these analyses would provide additional clarity, particularly for clinicians and researchers interested in how different demographics or intervention models impact outcomes.

5. Highlight Practical Implications for Healthcare: Since this review could inform clinical and policy decisions, adding a few thoughts on the practical implications could enhance its relevance. Outlining how the findings might translate to on-the-ground changes for clinicians or system planners could make this protocol even more compelling.

Recommendation: Major Revision

In sum, this protocol addresses an important topic with real-world relevance, and I believe it has significant potential. With some refinements to the scope and methodology, it will be even more robust and ready for publication. I look forward to seeing the revised version, as I think your findings will make a meaningful contribution to the field. Thank you again for sharing your work—it’s exciting to see research that could shape future directions in youth mental health care.

Response to comments from reviewer #1: Consistent with the internal evaluation from the editorial office, no changes have been made in response to the comments from Reviewer 1.

Reviewer #2: This paper presents a systematic review protocol aimed at comparing the efficacy of home-based versus inpatient treatment for children and adolescents experiencing psychiatric crises.

The protocol outlines a comprehensive methodology to evaluate treatment outcomes across multiple domains, including psychopathology, family functioning, and social functioning. It proposes to analyze both primary clinical outcomes and secondary parameters such as readmission rates and cost-effectiveness. The methodology includes a systematic search of major databases, rigorous quality assessment using standardized tools (RoB2 and ROBINS-I), and meta-analysis where appropriate.

1. The protocol's criteria for "psychiatric crisis" The protocol lacks precise clinical parameters for defining "psychiatric crisis," which could lead to inconsistencies in study selection and comparability. It is encouraged that have a brief of explain about psychiatric crisis in the introduction part.

2. Figure Clarity: The figures in the paper are unclear, making it difficult to discern specific details. It is recommended to upload high-resolution images.

3. Line 22 - Author Attribution: Only one author is marked, which may lead to misunderstandings regarding contributions.

Addressing these limitations will enhance the scientific rigor and practical applicability of the study's findings.

Response to comments from reviewer #2:

1. In the introduction, we have provided a detailed definition of the terms “psychiatric crisis” and “psychiatric emergency”. Given the ambiguity surrounding these terms in both the literature and clinical practice, we have provided a comprehensive definition of "psychiatric crisis" as it applies to our study. In the section entitled "Eligibility criteria," and more specifically in the subchapter designated "Population", the clinical parameters that are relevant for inclusion in the study have been described in exhaustive detail. These include specific psychiatric disorders and symptoms in the context of psychiatric crises according to ICD-10-CM, as well as the criteria for excluded diagnoses and symptoms. Furthermore, we have added that in clinical practice, an unequivocal assignment is often not possible due to the co-occurrence of internalizing and externalizing disorder patterns as comorbid presentations.

2. We have reattached Figure 1 with improved resolution and enhanced readability.

3. We have now fully listed the co-authors BB, GS, JS, and AN on the title page and specified their respective roles in the preparation of the manuscript.

Reviewer #3: This systematic review protocol aims to evaluate and compare the effectiveness and cost-effectiveness of home-based and inpatient treatments for children and adolescents in psychiatric crises. The authors propose a rigorous methodological approach, adhering to the PRISMA guidelines and incorporating a structured risk assessment using the Cochrane RoB2 and ROBINS-I tools. The study is valuable as it addresses a significant gap in understanding alternative psychiatric care for young individuals, offering potential insights into less restrictive, more family-centered treatment options.

Below are specific points to consider for enhancing the manuscript:

1. While the manuscript specifies the inclusion of children and adolescents under 18 from OECD countries, further clarity on the demographic and clinical characteristics (e.g., psychiatric diagnoses) that will be included would enhance reproducibility and relevance.

2. Clarifying the operational definitions for “home-based” interventions could strengthen the manuscript, as these can vary significantly across settings and countries.

3. Including a brief justification for the chosen follow-up intervals (e.g., 1, 3-6, and 6-12 months) and their relevance to long-term outcomes would enhance readers' understanding of these selections.

Response to comments from reviewer #3:

1. In the chapter "Eligibility Criteria", and more specifically in the subchapter titled "Population," we have specified the demographic and clinical characteristics in detail. This includes descriptions of the living conditions of the included population (e.g., exclusion if children and adolescents are in an institutional living arrangement). In response to Reviewer #2's question, we have detailed the included and excluded clinical parameters according to ICD-10-CM, noting that, in the case of comorbid disorders, multiple diagnoses/symptoms can occur simultaneously.

2. We have added a new chapter titled "Definition and Scope of Home-Based Models," to the manuscript to provide a clear definition of the term "home-based" within the context of this study. In the chapter entitled "Eligibility Criteria," and more specifically the subchapter entitled "Experimental Intervention," we outline the inclusion and exclusion criteria for home-based models as they apply to this research. Despite the clear delineation provided, heterogeneity among the models is anticipated when conducting an international comparison. This issue is further explored in the chapter entitled "Discussion."

3. We provided a brief justification for the chosen follow-up intervals in the chapter "Data Collection and Extraction."

Reviewer #4:

Major Comments:

1. Language could be streamlined for better readability. Consider rephrasing to avoid complex, multi-clause sentences that may hinder reader comprehension.

2. The scope of inclusion and exclusion criteria regarding mixed models and family-based intervention specifics could benefit from further clarification. For example, specifying clear distinctions between what constitutes "mixed interventions" versus standalone models would help ensure replicability of the protocol.

3. The decision to limit the population to OECD countries to reduce heterogeneity is commendable, but further justification would be more beneficial as some OECD countries may have different baseline healthcare structures and outcomes which could impact results.

4. Given the diversity of interventions included, the review might also benefit from incorporating additional bias assessments relevant to intervention studies. For example, the potential for performance bias, particularly in non-randomized studies where blinding may be challenging.

5. A brief overview of the decision-analytic framework and how cost data (particularly for various health systems within OECD countries) will be standardized or adjusted would be beneficial for clarity.

Minor Comments:

1. The manuscript occasionally uses varying terminology (e.g., “rehospitalization” vs. “readmission”). Consistent use of terminology throughout the text will improve clarity.

2. Lines 425–427: the choice of 10 studies as a minimum threshold could be briefly justified or referenced to support this decision.

Response to comments from reviewer #4:

Major Comments:

1. We have simplified and rephrased some complex sentences to enhance readability.

2. In the newly added chapter "Definition and Scope of Home-Based Models," we have further defined what is meant by "mixed models" and illustrated this with two models from England (SDS) and Germany (Hot-BITs). In the "Eligibility Criteria," more specifically in the subchapter "Experimental Intervention," we detail the precise inclusion and exclusion criteria for both model forms (mixed and stand-alone).

3. We have expanded the chapter "Discussion" to include a discussion on the expected heterogeneity of home-based models in an international context.

4. We examine publication bias using funnel plots and the Egger test in the chapter entitled „Meta-biases”. We discuss the challenge of a lack of blinding, which arises from the investigators and the study population being aware of the research site, in the domains of RoB 2 and ROBINS-I.

5. The study protocol is solely aimed at describing the procedure for the systematic literature search. The decision-analytic modeling will be discussed in a separate work. Therefore, we have clarified in the "Objectives" chapter that it is solely about identifying the transition probabilities for the model as part of the secondary outcomes. The costs and further details of the model are initially excluded.

Minor Comments:

1. We have standardized various terms such as "rehospitalization/readmission" and "hospitalization/ inpatient treatment" to ensure a more consistent terminology.

2. We have adjusted it and expanded to 20 studies, as the literature suggests that 20 studies are necessary for adequate power in the context of the Egger test.

Thank you again for considering this submission and I look forward to hearing from you and contributing to PLOS One.

Sincerely,

Karolina Foremnik

---

## [Decision Letter · Decision Letter 1]

22 Jan 2025

Efficacy of home-based and inpatient treatment for children and adolescents in psychiatric crisis: A systematic review protocol

PONE-D-24-45550R1

Dear Dr. Karolina Foremnik,

We’re pleased to inform you that your manuscript has been judged scientifically suitable for publication and will be formally accepted for publication once it meets all outstanding technical requirements.

Kind regards,

Cho-Hao Howard Lee, M.D.

Academic Editor

PLOS ONE

---

## [Editor Report · Acceptance letter]

PONE-D-24-45550R1

PLOS ONE

Dear Dr. Foremnik,

I'm pleased to inform you that your manuscript has been deemed suitable for publication in PLOS ONE. Congratulations! Your manuscript is now being handed over to our production team.

Kind regards,

on behalf of

Dr. Cho-Hao Howard Lee

Academic Editor

PLOS ONE